# Simulated Annealing-Based Optimization for the Coverage Path Planning of Multiple Unmanned Surface Vehicles in ECDIS

1st Yunwei Li
School of Marine Electrical Engineering
Dalian Maritime University
State Key Laboratory of Maritime
Technology and Safety
Dalian, China
ywli@dlmu.edu.cn

2nd Nan Gu
School of Marine Electrical Engineering
Dalian Maritime University
State Key Laboratory of Maritime
Technology and Safety
Dalian, China
ngu@dlmu.edu.cn

3rd Jiyang Jia
School of Marine Electrical Engineering
Dalian Maritime University
State Key Laboratory of Maritime
Technology and Safety
Dalian, China
jjy_081228@dlmu.edu.cn

4th Zhouhua Peng
School of Marine Electrical Engineering
Dalian Maritime University
State Key Laboratory of Maritime
Technology and Safety
Dalian, China
zhpeng@dlmu.edu.cn

5th Dan Wang
School of Marine Electrical Engineering
Dalian Maritime University
State Key Laboratory of Maritime
Technology and Safety
Dalian, China
dwang@dlmu.edu.cn

*Abstract*—This paper addresses the coverage path planning of multiple unmanned surface vehicles (USVs) based on electronic chart displays and information system. An optimized coverage path planning method is proposed to improve the coverage percentages and the coverage area of each USV leaves an additional degree of freedom for the operator to adjust further. Specifically, a grid representation method is proposed to fully cover the polygonal work area where three kinds of grids, namely, "Free", "Obstacle", and "Initial USV Position" are obtained. Next, a simulated annealing-based optimization method is proposed to optimize the placement of the grid by moving and rotating such that a more "Free" grid can be obtained. Then, an improved divide area based on the robot's initial position method is proposed to achieve proportional area division. Finally, the coverage paths are generated by using a spanning tree coverage method. Simulation results verify the effectiveness of the proposed simulated annealing-based optimization for the coverage path planning of multiple USVs.

*Index Terms*—Coverage path planning, unmanned surface vehicle, electronic chart display and information system, simulated annealing method.

## I. INTRODUCTION

Unmanned surface vehicle (USV), due to their characteristics such as compact size, lightweight construction, and auton-

This work was supported by the National Natural Science Foundation of China under Grant 52071044, and in part by the Doctoral Scientific Research Foundation of Liaoning Province under Grant 2024-BS-012, and in part by the Key Basic Research of Dalian under Grant 2023JJ11CG008, and in part by the Dalian High-level Talents Innovation Support Program under Grant 2022RQ010, and in part by the Bolian Research Funds of Dalian Maritime University and the Fundamental Research Funds for the Central Universities 3132024601, and in part by the Fundamental Research Funds for the Central Universities 3132023508. *(Corresponding author: Nan Gu; Zhouhua Peng).*

omy, serves as pivotal tools in the exploration and development of the oceans [1]–[4]. Multi-USV systems have a wider range of applications in ocean survey, sea rescue, marine scientific research, water patrol, and underwater equipment deployment compared to single USV [5]–[7]. Coverage path planning is a key component in the motion control of multiple USVs, which is to plan multiple paths such that an area can be fully covered.

Many coverage path planning methods are proposed in recent years. In [8]–[10], a coverage path planning method based on binary interval neighborhood network algorithm is proposed, which processes good adaptability and parallelism. In [11], [12], a coverage path planning method based on cell decomposition method is proposed, but this method needs to segment the map more precisely, which leads to higher computational complexity and reduces the path planning efficiency. In [13]–[16], a spanning tree coverage (STC) based coverage path planning method is proposed for multiple robots. The coverage path generated by this method does not repeat, but the search path depends on the initial position of each robot, which is prone to the problem of path overlap. In [17], an STC method based on auction and bidding process is presented, which balance the coverage path length of each robot. In [18], a divide areas based on robots initial positions (DARP) method is combined with STC method, which improves the coverage efficiency of multiple robots. However, each of these methods suffers from insufficient coverage or high computational requirements.

Based on the above discussions, this paper studies the coverage path planning of multiple USVs in electronic chart

display and information system (ECDIS). An optimized coverage path planning method is proposed to improve the coverage percentages and reduce the number of turns. First, based on three types of grid including "Free", "Obstacle", and "Initial USV Position", a grid map is established over a selected polygonal work area. Next, a simulated annealing optimization method is used to optimize the established grip map by moving and rotating the grids. Then, an improved DARP method is proposed to achieve the proportional area division and the coverage paths are generated by using a spanning tree coverage method. Finally, based on the ECDIS platform, simulation verifies the effectiveness of the proposed optimization based coverage path planning for multiple USVs.

## II. PROBLEM FORMULATION

ECDIS is a navigation information system with an electronic chart database, as well as navigational and piloting information. This information is required to navigate for USV, a multiple USVs coverage path planning optimization method is applied to ECDIS. Firstly, the initialization setup is performed. Next, the map model is transformed, optimized and solved. Finally, the coverage path for each USV is generated. The framework of the proposed optimized coverage path planning method of multiple USVs is shown in Fig. 1.

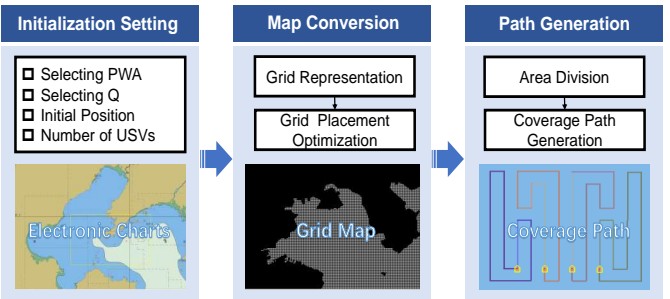

Fig. 1. Framework of the proposed optimized coverage path planning method of multiple USVs based on ECDIS.

Specifically, a polygonal work area (PWA) is selected manually in ECDIS, then obstacle areas can be selected in the selected PWA. PWA vertex coordinates and obstacle $Q$ coordinates are expressed in the format of the world geodetic system coordinate system 1984 (WGS84) , as shown in the following:

$$P\ WA = \{(X_1, Y_1), ...(X_n, Y_n)\}, \tag{1}$$

$$Q = \{\{(X_{o_1|1}, Y_{o_1|1}), ...(X_{o_1|p_1}, Y_{o_1|p_1})\}, \\ ...\{(X_{o_q|1}, Y_{o_q|1}), ...(X_{o_q|p_q}, Y_{o_q|p_q})\}\}, \tag{2}$$

where $X, Y$ represent latitude and longitude coordinates; $n$ is the number of polygon vertices; $o_i(i = 1, ...q)$ denotes the $i$th obstacle; $p_i(i = 3, ...q)$ is the number of vertices in each obstacle region, and $q$ is the number of obstacles.

The initial positions of the USVs in the PWA, which can also be represented in WGS84 format:

$$pos_{init} = \{(X_{s_1}, Y_{s_1}), ...(X_{s_m}, Y_{s_m}\}, \tag{3}$$

where $m$ is the USV number; $s_i(i = 1, ...m)$ denotes the $i$th USV. As an output of this method, a set of paths is generated, one path for each USV involved in the mission, thus providing complete coverage of the PWA:

$$path_{all} = \{\{(X_{z_1|1}, Y_{z1|1}), ...(X_{z_1|r_1}, Y_{z_1|r_1})\}, \\ ...\{(X_{z_m|1}, Y_{z_m|1}), ...(X_{z_m|r_t}, Y_{z_m|r_t})\}\}, \tag{4}$$

where $r_i(i = 1, ...t)$ is the number of waypoints per USV; $z_i(i = 1, ...m)$ denotes the $i$th path.

## III. COVERAGE PATH PLANNING DESIGN

This section presents simulated annealing-based optimization for the coverage path planning of multiple unmanned surface vehicles in ECDIS. It includes the grid representation, grid placement optimization, and area division and coverage path generation.

### A. Grid Representation

Traditional ECDIS-based USV path planning algorithms need to read the chart information of all points in the work area to create a grid map. In this paper, the optimization for coverage path planning based on simulated annealing involves extracting PWA vertex information and obstacle area vertex information. These are subsequently converted into a grid map based on the desired planning accuracy. This greatly decreases the computation time and lowers the requirement on computer configuration. After the conversion, a node is created at the center of each grid cell in grid map. Through the nodes, the minimum spanning tree is constructed, and eventually the coverage paths are generated with it. Considering different task requirements, set the planning accuracy in meters ($\delta$), i.e., half the distance between neighboring trajectories. Thus, the distance between two nodes is $\Delta = 2 \times \delta$.

To represent the PWA, creat a rectangular border that holds $a \times b$ grids of side length $\Delta$. As shown in Fig. 2 (red border):

$$a = \lfloor \frac{X_{\max} - X_{\min}}{\Delta} \rfloor, \tag{5}$$

$$b = \lfloor \frac{Y_{\max} - Y_{\min}}{\Delta} \rfloor, \tag{6}$$

and $\lfloor k \rfloor$ denotes the floor of $k$ (largest integer less than the given real number).

After that, grids are incorporated, where each grid center represents a node. These nodes can exist in three states: $Free$, $Obstacle$, and $Initial\ USVs'\ Position$. These states dictate their treatment in the path planning process, illustrated in Fig. 3 with green, black, and red colors respectively.

### B. Grid Placement Optimization

Limitations of the grid map-based path planning method include the inability of the grid map to fully select the planning area for representation, which leads to insufficient coverage of the planning area. To address this limitation, a common adopted solution is to generate more accurate grid maps by lowering the size of the accuracy, but it will also lead to a significant increase in time and cost to perform the tasks,

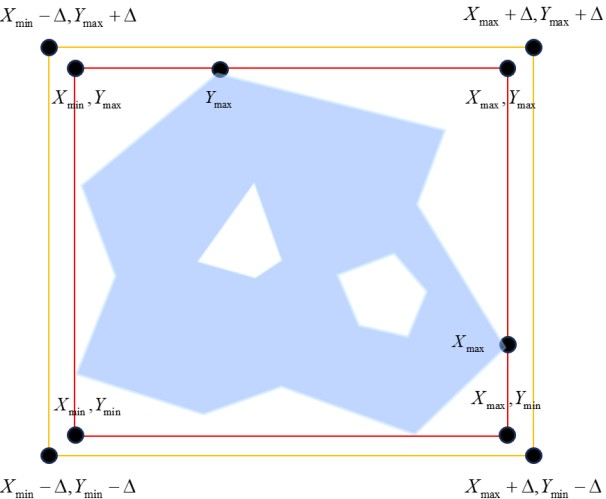

Fig. 2. Standard and augmented rectangular borders.

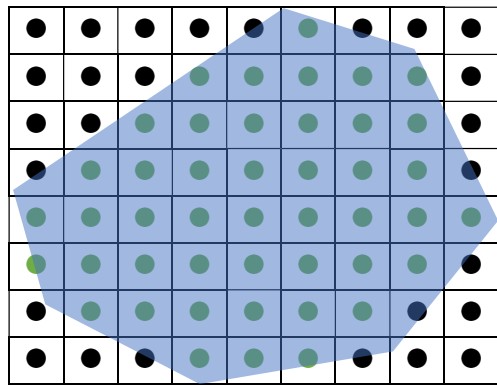

Fig. 4. Grid map before optimization.

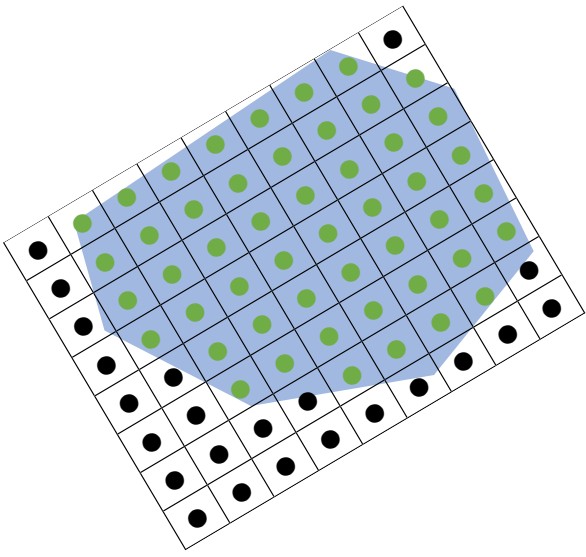

Fig. 3. Nodes are placed on polygons.

Fig. 5. Grid map after optimization.

and higher requirement on computer configuration. In this paper, an optimization method based on a simulated annealing algorithm is proposed to optimize the representation of grid maps.

The method is that by rotating and moving standard rectangular border, there will be more $Free$ nodes contained in PWA, which will effectively increase the overall coverage. For example, the number of $Free$ nodes before optimization is 45 shown in Fig. 4 , and after rotation and movement, the number of $Free$ nodes increases to 49 in Fig. 5. Based on this, optimization is performed using the simulated annealing algorithm to maximize the overall coverage of the PWA. Define the translation of the polygon's X-axis to be $C_x$, in a range $[0, \Delta]$, the translation of the polygon's Y-axis to be $C_y$, in a range $[0, \Delta]$, and the angle of rotation of the polygon to be $C_\theta$, in a range $[0, 90°]$.

Define an optimization index $F$ to denote the model associated with PWA coverage potential given a certain planning accuracy, and is expressed as:

$$F = k_1 \cdot F_1 + k_2 \cdot F_2 - k_3 \cdot F_3. \qquad (7)$$

$F$ consists of three independent normalized $F_i$ terms satisfying:

$$0 \le F_i \le 1 \in \Re, i \in \{1, 2, 3\}, \qquad (8)$$

The term $F_1$ denotes the basic objective of the whole optimization process, i.e., fitting the maximum number of nodes possible within a given PWA, defined as follows:

$$F_1 = \frac{\beta}{\frac{S_p}{\Delta^2}}, \qquad (9)$$

where $\beta$ represents the number of nodes placed within the polygon; $S_p$ represents the area of the polygon. The theoretical maximum number of nodes that can be accommodated within a given polygon is denoted as:

$$\beta \le \frac{S_p}{\Delta^2}, \qquad (10)$$

both $S_p$ and $\Delta$ are constants after region selection. Therefore, maximizing the $F_1$ term results in the maximal placement of nodes within the polygon.

In addition, the $F_2$ term provides better options for node placement within the polygon, increasing the coverage of the PWA edge region. The definition of $F_2$ is given as:

$$F_2 = \frac{S_p}{S_s}, \tag{11}$$

where $S_s$ represents the area of the enhanced bounding box. In the example in Fig. 2, $S_p$ represents the blue area of the polygon and $S_s$ represents the area of the yellow rectangular bounding box. Because PWA is constant, $S_s$ can be minimized by maximizing the overall $F_2$ term. The purpose of adding this term is mainly to control the rotation angle $C_\theta$ to optimize. as a result, the $F_2$ item is also normalized.

Finally, the $F_3$ term improves the location of nodes within a selected PWA, allowing for path alignment at the boundaries of the region and increasing the coverage of the range of the edge region. $F_3$ is defined as follows:

$$F_3 = \frac{||X_{\max_s} - X_{\max}| - |X_{\min} - X_{\min_s}||}{2 \cdot |X_{\max_s} - X_{\min_s}|} + \frac{||Y_{\max_s} - Y_{\max}| - |Y_{\min} - Y_{\min_s}||}{2 \cdot |Y_{\max_s} - Y_{\min_s}|}, \tag{12}$$

where $\{X_{\max_s}, X_{\min_s}, Y_{\max_s}, Y_{\min_s}\}$ are the vertices of the augmented rectangular border (yellow border in Fig. 2), and $\{X_{\max}, X_{\min}, Y_{\max}, Y_{\min}\}$ are the vertices of the standard rectangular border (red border in Fig. 2). The $F_3$ term is used to reconcile the absolute difference between the edge of the polygon and the augmented rectangular border in each axis, i.e., to fine-tune $C_x$ and $C_y$ so that the resulting paths are centered, and $F_3$ is normalized as well.

In order to optimize the node representation and placement for a given PWA and planning accuracy, a simulated annealing algorithm is used to find a solution of $(C_x, C_y, C_\theta)$ that maximizes $F$. $F_1$ and $F_2$ contribute positively to the overall optimization index and act as rewards, while $F_3$ acts as a penalty by appropriately reducing the $F_1$ and $F_2$ terms. $k_1$, $k_2$, and $k_3$ are constants to regulate each term, and they are allowed to take values in the following range:

$$0 \le k_1, k_2, k_3 \le 1 \in \Re, \quad k_1 + k_2 = 1. \tag{13}$$

An overview of the grid placement optimization algorithm is given in Algorithm 1, where $T$ is the initialization temperature, $T_{\min}$ is the minimum temperature, $\alpha$ is the temperature drop rate, $K$ is the number of iterations at each temperature, $\gamma$ is a constant, $P$ is the acceptance probability of solution, and the algorithm returns the global optimal solution $F_{max}$ and $C_x$, $C_y$, $C_\theta$ when the optimal solution is reached.

### C. Area Division and Coverage Path Generation

In this section, the DARP algorithm is first improved to achieve proportional area division. The divided area is then combined with the STC algorithm to provide an independent path for each USV.

---

**Algorithm 1** Grid placement optimization

**Input:** $T$, $T_{\min}$, $\alpha$, $K$, $\delta$, $k_1$, $k_2$, $k_3$, $\gamma$
**Output:** $F_{\max}$, $C_x$, $C_y$, $C_\theta$
1: **Initialize:** $C_x = \delta$; $C_y = \delta$; $C_\theta = 0$; $F_{\max} = 0$
2: **while** $T > T_{\min}$ **do**
3:    **while** $(i <= K - 1)$ **do**
4:      $F_{cur} = k_1 \cdot F_1 + k_2 \cdot F_2 - k_3 \cdot F_3$; // Get the current solution's optimization index
5:      **if** $F_{cur} > F_{\max}$ **then**
6:        $F_{\max} = F_{cur}$;// Update the optimization index
7:      **end if**
8:      **if** $(i = K - 1)$ **then**
9:        Create a new state for new solution
10:        $C_{x_{new}} = C_{x_{cur}} + random(0, 1) \times 2 \times \delta$;
11:        $C_{y_{new}} = C_{y_{cur}} + random(0, 1) \times 2 \times \delta$;
12:        $C_{\theta_{new}} = C_{\theta_{cur}} + random(0, 1) \times 90$;
13:      **else**
14:        Create a neighboring state for new solution
15:        **if** $(random(0, 1) > \gamma)$ **then**
16:          $C_{x_{new}} = C_{x_{cur}} + random(0, 1) \times \delta/4$;
17:        **else**
18:          $C_{x_{new}} = C_{x_{cur}} + random(0, 1) \times \delta/(-4)$;
19:        **end if**
20:        **if** $(random(0, 1) > \gamma)$ **then**
21:          $C_{y_{new}} = C_{y_{cur}} + random(0, 1) \times \delta/4$;
22:        **else**
23:          $C_{y_{new}} = C_{y_{cur}} + random(0, 1) \times \delta/(-4)$;
24:        **end if**
25:        **if** $(random(0, 1) > \gamma)$ **then**
26:          $C_{\theta_{new}} = C_{\theta_{cur}} + random(0, 1) \times 45$;
27:        **else**
28:          $C_{\theta_{new}} = C_{\theta_{cur}} + random(0, 1) \times (-45)$;
29:        **end if**
30:      **end if**
31:      $P = e^{(F_{cur} - F_{new})/T}$; // Calculate the acceptance probability
32:      **if** $P > random(0, 1)$ **then**
33:        $F_{max} = F_{new}$; // Update optimization index
34:      **end if**
35:    **end while**
36:    $T = T \times \alpha$; // Update temperature
37: **end while**

---

The traditional DARP method provides equal area partition, which means that all USVs will be assigned exactly the same proportion of area. In this work, DARP is extended and now capable to perform proportional area division, where the percentage is defined by the user, facilitating the simultaneous operation of heterogeneous USVs with different energies in the same task:

$$\sum_{i=1}^{m} G_i = 1, \tag{14}$$

where $G_i$ denotes the percentage of regions of the $i$th USV.

Each sub-region is generated to cover the path after completing the task area division. In this paper, the STC method is used as the basic method for generating coverage path planning. The principle of the STC method is to generate a minimum spanning trees (MSTs) in each subregion and then generate paths around each MST.

## IV. SIMULATIONS

In this section, simulation results are presented to illustrate the effectiveness of the simulated annealing based optimized coverage path planning method for multiple USVs.

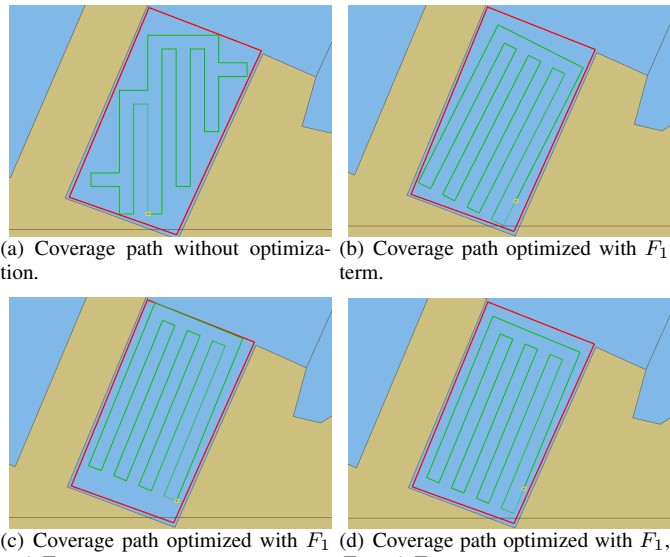

(a) Coverage path without optimization.

(b) Coverage path optimized with $F_1$ term.

(c) Coverage path optimized with $F_1$ and $F_2$ terms.

(d) Coverage path optimized with $F_1$, $F_2$ and $F_3$ terms.

Fig. 6. Node placement optimizes coverage paths.

TABLE I
OPTIMIZATION PARAMETER RESULT ANALYSIS.

|  | Non-Optimized | F1 | F1+F2 | **F1+F2+F3** |
|---|---|---|---|---|
| Poc(%) | 76.35 | 81.36 | 83.67 | **91.42** |
| Turns | 26 | 16 | 16 | **16** |
| N-Turns | 0.13 | 0.08 | 0.08 | **0.08** |
| Length(km) | 3.84 | 4.48 | 4.48 | **4.48** |
| N-Length | 19.59 | 22.85 | 22.86 | **22.86** |

In the simulation, a part of Dalian harbor is selected in ECDIS as PWA. In order to validate the effectiveness of the method, the number of USV is selected as 1, and planning accuracy as 20m. The simulation results can be seen in Fig. 6. Fig. 6 (a) shows the coverage path without optimization; Fig. 6 (b) shows the optimization results under $F_1$ term; Fig. 6 (c) shows the optimization results under $F_1$ term and $F_2$ term; Fig. 6 (d) shows the optimization results under $F_1$ term, $F_2$ term and $F_3$ term. Table I compares five criteria under different methods. It shows that the proposed method is able to increase percentage of coverage (POC), path length, and normalized path length (N-Length) $(\frac{m}{1000m^2})$, as well as decrease in number of turns and normalized number of turns (N-Turns) $(\frac{Turns}{1000m^2})$. Specifically, the number of turns without

optimization is 26, the coverage is 76.35%, and the path length is 3.84 km. After the introduction of the $F_1$ term the number of turns is 16, the coverage is 81.36%, and the path length is 4.48 km. It can be seen that the number of turns is significantly decreased and the coverage is significantly improved. After the introduction of the $F_1$ term and the $F_2$ term, the number of turns is 16, the coverage is 83.67% and the path length is 4.48 km. $F_2$ term is mainly used for the rotation of the generated path to obtain the optimal $C_\theta$. After the introduction of the $F_1$, $F_2$ and $F_3$, the number of turns is 16, the coverage rate is 91.42%, and the path length is 4.8 km. $F_3$ term makes the generated path centrally placed, which is more conducive to the coverage of the edge region.

Therefore, the introduction of the optimization index $F$ has a major improvement on both the number of turns and the coverage. Apart from that, planning accuracy is also a key factor affecting the coverage. For the same PWA, optimized coverage can reach more than 97% when the planning accuracy is 10 meters, at the expense of more computing time.

In addition, in order to verify whether the number of USVs has impact on the effectiveness of the optimization method, and also to verify the planning effect of proportional area division. 20km$^2$ sea area near West Ant Island of Dalian is selected as PWA, and the planning accuracy is defined as 50 meters. The coverage paths are planned for the number of USVs as 5, 10, 20, and 50 respectively. The planning results for 5 USVs and 50 USVs are shown in Fig. 8 and Fig. 9, where the red area is the PWA and the yellow areas are the obstacle areas. The area division proportions of the 5 USVs in Fig. 7 are set to 10%, 10%, 20%, 20%, 40%. Table II indicates optimized data for different number of USVs. The results show that the number of USVs has no significant effect on the coverage of the generated paths, while the algorithm running time rises significantly when USVs number reaches 50.

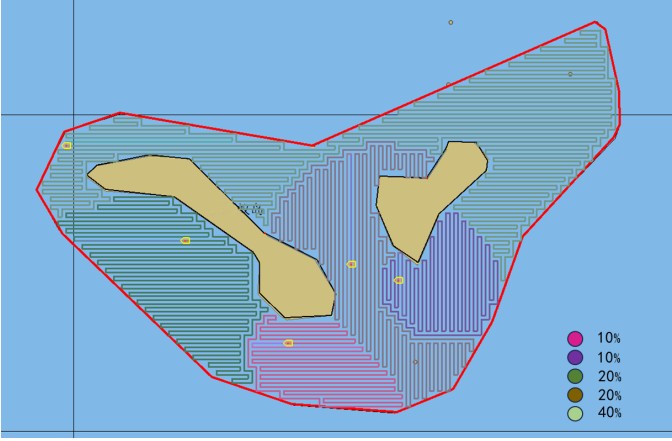

| | |
|---|---|
| 🔴 | 10% |
| 🟣 | 10% |
| 🟦 | 20% |
| 🟤 | 20% |
| ⚪ | 40% |

Fig. 7. Coverage paths for 5 USVs assigned by proportional area division.

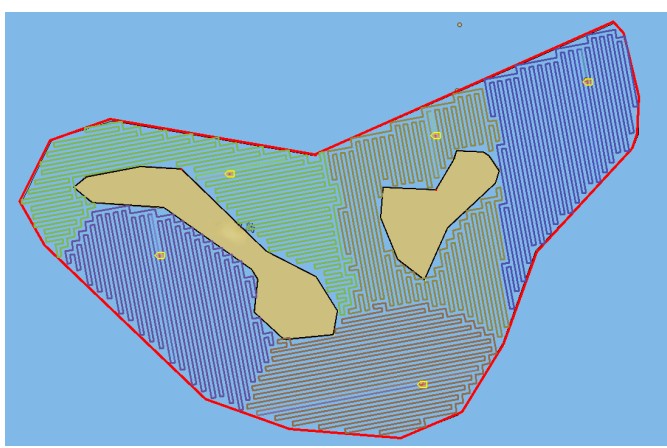

Fig. 8. Coverage paths for 5 USVs.

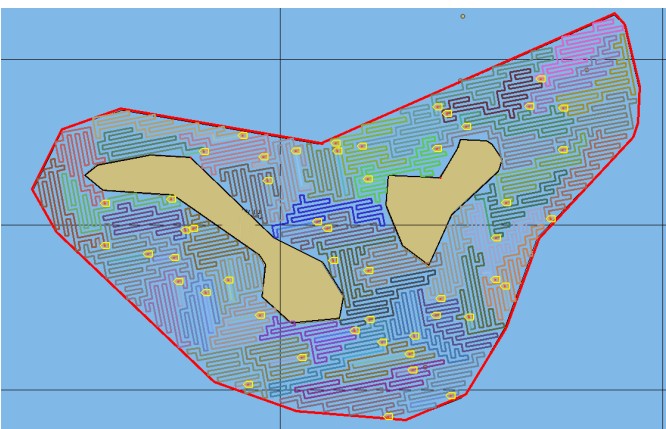

Fig. 9. Coverage paths for 50 USVs.

TABLE II
OPTIMIZED DATA FOR DIFFERENT NUMBER OF USVS.

| Number of USVs | Poc(%) | Turns | Length(km) | Time(s) |
|---|---|---|---|---|
| 5 | 97.68 | 839 | 33.34 | 91.70 |
| 10 | 97.63 | 969 | 33.33 | 93.71 |
| 20 | 97.71 | 1218 | 33.53 | 96.39 |
| 50 | 97.56 | 1924 | 33.02 | 908.70 |

## V. CONCLUSION

This paper investigates the coverage path planning of multiple USVs in ECDIS. A simulated annealing-based optimization for the coverage path planning of multiple USVs is proposed. Simulations of multiple target sea areas are conducted to verify the effectiveness of the proposed coverage path planning method for increasing coverage percentages, reducing the number of turns, and increasing the operator's degree of freedom. The simulated annealing-based optimization for the coverage path planning of multiple USVs in ECDIS proposed herein may provide a feasible method for future large-scale maritime coverage operations of multiple USVs.

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
