# OpenReview forum: "Simulated Annealing-Based Optimization for the Coverage Path Planning of Multiple Unmanned Surface Vehicles in ECDIS"
_IEEE.org/ICIST/2024/Conference — IEEE ICIST 2024 Conference Submission_

### Official Review · Reviewer_EkXN · 2024-08-22
**All in all good, but there are still some small problems.**

**Rating:** 9
**Confidence:** 5

**Review:**

This paper discusses a simulated annealing-based optimization method for coverage path planning of multiple unmanned surface vehicles (USVs) within an electronic chart display and information system (ECDIS). The study proposes an optimized coverage path planning approach that enhances coverage percentages, reduces the number of turns, and provides operators with increased flexibility. The method involves grid representation, grid placement optimization, area division, and coverage path generation. Simulation results demonstrate the effectiveness of this approach, showing significant improvements in coverage rates and path efficiency across various maritime scenarios. On the whole, the paper is good, but there are still some problems that need to be corrected.

Comments:

1.The structure of the article is generally clear, but it may benefit from a more distinct separation between the background introduction and the discussion of existing methods in the introduction section, making it easier for readers to identify the novelty of the study.

2.The overall language is fluent, but some sections with technical jargon might be simplified to improve readability, particularly in the more detailed technical discussions.

3.The research content is well constructed. But there are grammatical errors and incorrect English writing in the current manuscript. Please make the necessary corrections.

---

### Official Review · Reviewer_rfAJ · 2024-08-22
**Simulated Annealing-Based Optimization for the Coverage Path Planning of Multiple Unmanned Surface Vehicles in ECDIS**

**Rating:** 7
**Confidence:** 4

**Review:**

This paper addresses the coverage path planning of multiple unmanned surface vehicles (USVs) based on electronic chart displays and information system. This work is well organized. Below are some comments.
1.	The innovation is not clear. It is suggested that the authors restate the innovation and contributions.
2.	The careful proofreading should be further accomplished.

---

### Official Review · Reviewer_SH9q · 2024-08-22
**Accept**

**Rating:** 8
**Confidence:** 4

**Review:**

This paper focuses on the coverage path planning of multiple unmanned surface vehicles and mainly includes the study of an optimized coverage path planning method and a simulated annealing-based optimization method. This paper gives details of the design process for the grid representation, grid placement optimization, and area division and coverage path generation, which improves the coverage of each USV and adds additional adjustment degrees of freedom. Simulation results verify the effectiveness of the proposed simulated annealing-based optimization for the coverage path planning of multiple USVs. But there are still two suggestions:
1. The innovation of this paper can be clearly expressed in Section Ⅰ;
2. The superiority of this paper to other published articles can be demonstrated in simulation.

---

### Official Review · Reviewer_uDi1 · 2024-08-27
**The topic under consideration is interesting. This paper can be accepted after minor modifications.**

**Rating:** 9
**Confidence:** 3

**Review:**

This paper investigates the coverage path planning of multiple unmanned surface vehicles based on electronic chart displays and information system. The topic under consideration is interesting. Detailed comments and suggestions are listed as follows.
1.	The English writing of the paper needs to be further polished, and some typos should be fixed.
2.	In (5) and (6), $X_{max}$, $ X_{min}$, $Y_{max}$ and $Y_{min}$ should be defined detailedly.
3.	 Suggest adding the latest references. In addition, the manuscript format of references should be standardized.

---

### Decision · Program_Chairs · 2024-09-06

Accept (Oral)